# Generation Healthy Kids: Protocol for a cluster-randomized controlled trial of a multi-component and multi-setting intervention to promote healthy weight and wellbeing in 6–11-year-old children in Denmark

**Louise T. Thomsen**[1,2], **Jesper Schmidt-Persson**[3], **Camilla Trab Damsgaard**[4], **Peter Krustrup**[3], **Anders Grøntved**[3], **Rikke Fredenslund Krølner**[5], **Glen Nielsen**[4], **Jesper Lundbye-Jensen**[4], **Thomas Skovgaard**[3], **Christian Mølgaard**[4], **Anders Blædel Gottlieb Hansen**[1], **Didde Hoeeg**[1,2], **Malte Nejst Larsen**[3], **Line Lund**[5], **Paulina Sander Melby**[3], **Natascha Holbæk Pedersen**[3], **Jens Troelsen**[3], **Nikolai Baastrup Nordsborg**[4], **Ulla Toft**[1,2,6] *

**1** Center for Clinical Research and Prevention, Bispebjerg-Frederiksberg Hospital, Frederiksberg, Denmark, **2** Department of Prevention, Health Promotion and Community Care, Steno Diabetes Center Copenhagen, Herlev, Denmark, **3** Department of Sports Science and Clinical Biomechanics, University of Southern Denmark, Odense, Denmark, **4** Department of Nutrition, Exercise and Sports, University of Copenhagen, Copenhagen, Denmark, **5** National Institute of Public Health, University of Southern Denmark, Copenhagen, Denmark, **6** Section of Social Medicine, Department of Public Health, University of Copenhagen, Copenhagen, Denmark

* ulla.toft@regionh.dk

## Abstract

### Background

Childhood obesity can have significant negative consequences for children's wellbeing and long-term health. Prior school-based interventions to prevent child overweight and obesity have shown limited effects, highlighting the necessity for comprehensive approaches addressing complex drivers of childhood obesity. *"Generation Healthy Kids"* (GHK) is a multi-setting, multi-component intervention to promote healthy weight development, health and wellbeing in Danish children aged 6–11 years. This protocol describes the GHK main trial, which is a cluster-randomized trial evaluating effectiveness and implementation of the GHK intervention.

### Methods

Twenty-four schools from the Capital, Zealand and Southern Denmark Regions are randomly allocated 1:1 to intervention or control. The intervention will run for two school years (18–20 months) from October 2023 to June 2025 and will include children in 1st–3rd grade (approx. n = 1,600). The intervention targets multiple settings, including families, schools, after-school clubs, and local communities. Within four focus areas–diet, physical activity, screen media use, and sleep habits–the intervention incorporates several fixed elements, including a school lunch program and three weekly sessions of physical activity at school.

**Data Availability Statement:** No datasets were generated or analysed during the current study, as this is a study protocol. Since the trial data set will contain potentially identifying sensitive participant information, the full data set will not be publicly shared. When data collection is complete, and the main study findings have been published, a deidentified copy of the dataset may be shared with external researchers upon reasonable request. A motivated application must be sent to the Department of Nutrition, Exercise and Sports, Copenhagen University (contact via nexs@nexs.ku.dk). The applicant must demonstrate compliance with all ethical and data protection requirements before data can be shared.

**Funding:** The study is funded by the Novo Nordisk Foundation (www.novonordiskfonden.dk) by grant no. NNF22SA0077224 awarded to NBN (primary investigator), JT (co-primary investigator), UT, CTD, PK, AG, RFK, JLJ, GN, TS and CM. The protocol underwent external peer review. The funders played no role in design of the study, writing of this study protocol, nor in the decision to submit the protocol for publication. Furthermore, the funder will play no role in data collection, data analysis, interpretation of results nor in any decision to submit future publications from the study.

**Competing interests:** The authors have declared that no competing interests exist.

**Abbreviations:** CAMSA, Canadian Exercise and Movement Skills Ability test; FFM, Fat free mass; FM, Fat mass; GDPR, General Data Protection Regulation; GHK, Generation Healthy Kids; LCPG, Local Community Partnership Group; PI, Primary Investigator; STICKE, Systems Thinking in Community Knowledge Exchange; WP, Work Package.

Furthermore, building on whole-systems thinking, the intervention encompasses co-created elements developed in collaboration with local stakeholders, e.g. municipalities, sports clubs and supermarkets. This part of the intervention emphasizes building local capacity and engagement to promote child health. Effectiveness data will be collected from participating children and families at baseline, and at the end of school year one (after 6–8 months) and school year two (after 18–20 months). The primary outcome is the change in fat mass, measured by air-displacement plethysmography, from baseline to end-of-study in the intervention group compared to the control group. This is supplemented with numerous secondary outcomes and other prespecified outcomes related to child health and wellbeing. Furthermore, thorough process evaluation will be performed.

## Discussion

GHK combines evidence-based intervention elements targeting multiple settings with a whole-systems approach focusing on capacity building and stakeholder involvement. This novel approach holds promise as an innovative way to promote child health and wellbeing and prevent childhood obesity.

## Trial registration

ClinicalTrials.gov: NCT05940675 (registered on 4 July 2023).

## Background

Childhood overweight and obesity is a global challenge affecting more than 340 million children and adolescents worldwide [1]. The prevalence has increased considerably in the past decades, and childhood obesity is characterized by the World Health Organization as one of the most serious public health challenges for the 21st century [2]. According to the Lancet 2019 Commission, obesity can be considered a global epidemic that no country has been able to successfully reverse [3]. In Denmark, 12–13% of children aged 6–7 years have overweight or obesity [4], while the prevalence increases to 18–19% in children aged 14–15 years [4] and 53% in the adult population [5].

Childhood overweight and obesity can have significant negative consequences for children's wellbeing and long-term physical health. Children with obesity have increased risk of experiencing low quality of life, reduced self-esteem, and bullying or stigmatization [6–8]. Furthermore, children with overweight and obesity are more likely to have overweight or obesity as adults [9], thereby increasing their risk of several severe health conditions, including cardiovascular disease, diabetes, some cancer types, asthma, musculoskeletal conditions, and premature mortality [10].

The burden of childhood overweight and obesity is unevenly distributed between socioeconomic groups, both in Denmark and other high-income countries. Thus, the prevalence is generally higher in socioeconomically disadvantaged groups and families living in deprived areas, thereby contributing substantially to overall health inequality in many countries [11, 12]. The prevalence of overweight and obesity among Danish children aged 14–15 years is almost 2.5 times higher in low-income compared with high-income families, and 3-fold higher in households where parents have shorter compared with longer education [4].

## Prior interventions to prevent childhood overweight and obesity

Previous interventions to prevent childhood overweight and obesity focused mainly on dietary and/or physical activity interventions in schools [13–19]. The prior interventions generally showed no or only small effects on weight development [13–19], and the evidence of long-term effects is limited [14, 20]. It is therefore increasingly argued that preventing childhood overweight and obesity requires multi-component and multi-setting interventions, which focus not only on individual health behaviors, but also on changing the structural environments in which children and their families live [21–23]. Some earlier studies have shown an effect on overweight measures, and these were generally characterized by not only targeting the school setting, but also other settings where children spend their daily lives, e.g. both the family, school, and local community [18, 21, 24, 25]. Furthermore, interventions appear more likely to be effective if they target multiple risk factors for unhealthy weight [16, 18, 19], combine educational and structural intervention components [15, 19, 21] and actively involve parents [18, 24].

## Systems thinking as a framework for preventing childhood overweight and obesity

Recently, whole systems-thinking has been proposed as a useful framework for efforts to prevent childhood overweight and obesity [3, 23, 26–28]. Systems approaches view childhood obesity as a complex public health problem caused by a dynamic interplay between multiple factors at both genetic, behavioral, social, cultural, environmental, economic, and political level [3, 22, 27, 29]. Thus, these approaches explicitly address the systemic drivers of childhood obesity, while emphasizing the complex and non-linear relationships between many multilevel causal factors. Furthermore, whole systems-approaches stress the importance of building local capacity and involving multiple stakeholders in identifying, developing, and implementing interventions to enable sustainable and long-term systems change [23, 26]. So far relatively few studies have applied a whole-systems approach to childhood obesity prevention [23, 28], and to our knowledge only one of these prior studies used a randomized controlled study design [26]. In a Danish context, systems thinking is increasingly gaining acceptance as a promising approach to child obesity prevention and health promotion, but experience still remains limited [30, 31].

Considering the seriousness of the global obesity epidemic and the limited effects of prior preventive interventions, there is a strong need to develop and evaluate large-scale, multi-component preventive programs, which build upon existing evidence while simultaneously incorporating new and innovative approaches. Therefore, we initiated and developed the Generation Healthy Kids (GHK) multi-component intervention to meet the need for effective, comprehensive and sustainable interventions to promote healthy weight development and wellbeing in Danish children.

## The Generation Healthy Kids intervention

The overarching purpose of the GHK intervention is to promote healthy weight development and wellbeing in Danish children aged 6–11 years. The GHK intervention utilizes a population-based primary prevention approach intended to benefit the health and wellbeing of all children, not only those at high risk of having overweight or obesity [32]. The intervention is based on an ecological understanding of health behavior and health promotion, emphasizing the importance of addressing multiple determinants at both individual, interpersonal, organizational and community levels and targeting structural determinants of health [33]. From a

health equity perspective, the intervention aims to reduce social inequalities in childhood overweight and obesity. Thus, to reach children and families with low socioeconomic status, the GHK program emphasizes intervention components which aim to create healthier environments both at school and during leisure time, thereby making healthy choices easier for families and children.

The integrated GHK intervention focuses on four behavioral components influencing childhood overweight and obesity, which are at the same time important for overall child health and wellbeing: diet, physical activity, screen media use, and sleep habits [22, 34–37].

- Diet: Healthy diet and a balanced energy intake are crucial for children's overall growth, weight development, cardiometabolic health, cognitive function, and psychosocial wellbeing. According to national Danish dietary surveys [38], Danish children generally have lower intake of fruit and vegetables, wholegrains and fish, and higher intake of sugar, fat and salt than recommended by Danish and Nordic evidence-based dietary guidelines and recommendations [39, 40]. For example, it has been estimated that only 6% and 23% of Danish children, respectively, comply with recommendations for intake of fish and fruits/vegetables [38]. In addition, there is marked socioeconomic inequality in children's dietary patterns, with unhealthy diets being more prevalent in children of parents with shorter compared with longer education [38].

- Physical activity: Physical inactivity is a central risk factor for overweight and obesity and is associated with increased risk of multiple non-communicable diseases including diabetes, cardiovascular disease, and cancer [41]. For children, being physically active is associated with improved physical fitness, motor competences, cardiometabolic health, and cognitive performance as well as increased wellbeing [42–44]. The World Health Organization recommends that children should participate in at least 60 minutes of moderate-to-vigorous physical activity per day, and that vigorous intensity aerobic activities and activities strengthening muscle and bone should be incorporated at least three times per week [35]. However, in Denmark a recent study showed that only 26% of children aged 11–15 years meet these targets [45], and similar trends are seen in other countries [46, 47].

- Recreational screen media use: Children spend large amounts of their leisure time using screen media devices [48], and high screen media use has been associated with obesity, unhealthy eating, and depressive symptoms [36]. Altering recreational screen media habits in families has previously been associated with large increases in daily physical activity in children [49].

- Sleep habits: Insufficient sleep duration is recognized as an important risk factor for childhood overweight and obesity, which is hypothesized to be caused by altered hormone regulation leading to increased appetite and calorie intake; increased sedentary time; and decreased physical activity [37, 50]. Sleep of sufficient duration and quality is furthermore central for children's wellbeing and neurocognitive development [51]. Nevertheless, national surveys show that up to 10% of Danish 11-year-old children sleep less than the recommended hours per night [52], and similar patterns are seen in other European countries [53].

In the integrated multi-component GHK program, predefined intervention elements targeting diet, physical activity, screen media use and sleep will be combined with local capacity building and co-creation of interventions with national and local stakeholders using a whole-systems approach [23, 27]. The GHK intervention settings, components and stakeholders are further described in the Methods section. To achieve synergy and coherence between the

various intervention components, we will apply the Supersetting approach [54, 55] which stresses the importance of integrating and coordinating interventions at many levels in a local community, as well as the importance of engaging multiple local stakeholders and adapting interventions to the local context [55].

This protocol describes the GHK main trial, which is a cluster-randomized school- and community trial designed to investigate the effects of the multi-setting, multi-component GHK intervention on weight development, health, and wellbeing in Danish school children. The trial is planned to run for two school years at 24 schools (12 intervention and 12 control schools), and it will include children in 1st and 2nd grade at study start (aged 6–9 years at study start).

## Intervention development and the GHK pilot study

The development of the GHK intervention was inspired by the United Kingdom Medical Research Council's framework for developing and evaluating complex interventions [56], building upon pre-existing evidence, theory, context knowledge, input from relevant stakeholders, and a pilot study.

Prior to study start (July–November 2022), central national stakeholders within child health and wellbeing were involved in mapping existing obesity prevention initiatives and suggesting possible new actions to prevent childhood obesity in local communities in Denmark. The purposes of involving national stakeholders were 1) to ensure that the GHK intervention was anchored in a shared understanding of the underlying drivers of child overweight and obesity in Denmark; 2) to map and build upon existing successful national initiatives when designing GHK; and 3) to facilitate access to potential local collaborators in intervention communities, e.g. collaboration with local supermarkets supported by the chain national office. The involved stakeholders included actors from the public (e.g. Danish Health Authority, Danish Veterinary and Food Administration, several municipality representatives), private (e.g. a large Danish food retail chain), and non-profit sectors (e.g. Danish Sports Association, Danish Parent Organization, Center for Digital Youth Care, Danish Cancer Society etc.). The process will be further described in a separate publication, but briefly, it included an initial stakeholder analysis, a series of bilateral meetings, and two large workshops facilitated using Group Model Building [29, 57] and Systems Thinking in Community Knowledge Exchange (STICKE) software [58] to generate a system map of central drivers of child overweight and obesity.

Using inputs from the national workshops, coupled with preexisting evidence and experience in our research group on implementing school meals [59], physical activity in schools [60], and screen media interventions [49], a pilot version of the intervention was developed. Subsequently, the intervention was tested in a pilot- and feasibility study conducted from December 2022–May 2023 among five 1st and 2nd grade classes at one Danish public school in the Eastern part of Denmark (n = 81 children). The pilot study investigated the feasibility and acceptability of the planned intervention components and biomedical measurement schedule, based on observations, interviews, logbooks and questionnaires from school staff, school management, and parents. The findings of the pilot study, including the consequent adaptions of the intervention and measurement protocol, will be reported separately.

## Specific aims and hypotheses of the GHK main trial

The primary aim of the GHK main trial is to investigate the effectiveness of the GHK intervention on promoting healthy body composition as measured by fat mass (FM) in the intervention group compared with the control group. We hypothesize that the intervention will result

in less FM gain in the intervention group compared with the control group during the study period.

Secondary aims of the trial include:

1. To investigate the intervention's effect on numerous measures of child health and wellbeing, including other body composition measures; anthropometric measures and growth; child- and parental reported wellbeing; cognitive function; school performance and school absence; cardiometabolic health; physical fitness and motor function; physical activity levels and physical literacy; nutritional status, dietary intake, and food literacy; and screen media use and sleep habits.

2. To assess differential effects of the intervention according to sociodemographic, socioeconomic, genetic and other baseline characteristics, including investigating whether the intervention can reduce social inequality in the outcomes.

3. To evaluate implementation of the intervention, causal mechanisms and contextual factors which shape the intervention outcomes, as well as the engagement, ownership and capacity of national and local stakeholders in the development and implementation of the intervention.

4. To evaluate the effect of the intervention on the local food environment, including sales of healthy foods in supermarkets, restaurants, and fast-food outlets.

5. Finally, data collected in the study will be used to investigate and validate associations between behavioral measurements (e.g. dietary intake, physical activity, sleep and screen media habits) and outcomes related to health and wellbeing (e.g. body composition, cardiometabolic health, cognitive performance), with data analyzed as observational data both cross-sectionally and longitudinally.

## Design and methods

### Study design

The GHK trial is a parallel group cluster-randomized superiority trial with two arms. Fig 1 provides the schedule of enrolment, interventions, and assessments according to the 2013 SPIRIT statement, and Fig 2 provides an overview of the study design. A total of 24 schools (i.e., clusters) are randomly allocated to either intervention (the GHK multi-component intervention program) or control (no intervention). Children in 1st and 2nd grade at study start at the included schools are eligible to participate. The intervention runs for two school years. Baseline data are collected before initiation of the intervention (at the start of school year one), and outcomes will be assessed at the end of the first school year (6–8 months after baseline), and at the end of the second school year (18–20 months after baseline) (Figs 1 and 2).

### Sample size determination

The *à priori* target was to recruit 24 schools for the study. This target was based on sample size calculations using our primary outcome, i.e. FM in kg at 18–20 months after baseline. We expected to have a total of 2,350 eligible children in 1st and 2nd grade, corresponding to a mean of four classes per school and 24–25 students per class in 24 schools. Furthermore, we expected to enroll 1,645 children at baseline (participation rate of ≈70%), and a drop-out/missing data rate of 15%, resulting in an expected mean of 58 children in each of the 24 clusters in the analytical sample (n≈1,398).

| | PRE-BASELINE | | | | | | | | | | BASELINE | POST-BASELINE | | | | | | | | | | | | | | | CLOSE-OUT |
|---|---|---|---|---|---|---|---|---|---|---|---|---|---|---|---|---|---|---|---|---|---|---|---|---|---|---|---|
| | | | | | | | | | | | Baseline assessment (T0) | | | | | 1st follow-up assessment (T1) | | | | | | | | | | | 2nd follow-up assessment (T2) |
| Timepoint (months since baseline) | -10 | -9 | -8 | -7 | -6 | -5 | -4 | -3 | -2 | -1 | 0 | 1 | 2 | 3 | 4 | 5 | 6-8 | 9 | 10 | 11 | 12 | 13 | 14 | 15 | 16 | 17 | 18-20 |
| **ENROLMENT** | | | | | | | | | | | | | | | | | | | | | | | | | | | |
| Invitation and inclusion of schools (i.e., clusters)[a] | ◆━━━━━━━━◆ | | | | | | | | | | | | | | | | | | | | | | | | | | |
| Random allocation of clusters[a] | | | X | | X | | | | | | | | | | | | | | | | | | | | | | |
| Recruitment of individual study participants (i.e., children)[b] | | | | | | | | ◆━━◆ | | | | ◆━◆ | | | | | ◆━━━━◆ | | | | | | | | | | |
| Written informed parental consent[b] | | | | | | | | ◆━━◆ | | | | ◆━◆ | | | | | ◆━━━━◆ | | | | | | | | | | |
| **INTERVENTION AND CONTROL** | | | | | | | | | | | | | | | | | | | | | | | | | | | |
| Generation Healthy Kids intervention program | | | | | | | | | | | | ◆━━━━━━━━━━━━━━━━━━━━━━━━━━◆ | | | | | | | | | | | | | | | |
| Control (no intervention) | | | | | | | | | | | | ◆━━━━━━━━━━━━━━━━━━━━━━━━━━◆ | | | | | | | | | | | | | | | |
| **ASSESSMENTS[c]** | | | | | | | | | | | | | | | | | | | | | | | | | | | |
| **Body composition, anthropometry and cardiometabolic health** | | | | | | | | | | | | | | | | | | | | | | | | | | | |
| Body composition (air-displacement plethysmography) | | | | | | | | | | | X | | | | | | | | | | | | | | | | X |
| Body composition (bioimpedance) | | | | | | | | | | | X | | | | | | X | | | | | | | | | | X |
| Height, weight and waist circumference | | | | | | | | | | | X | | | | | | X | | | | | | | | | | X |
| Resting blood pressure and heart rate | | | | | | | | | | | X | | | | | | X | | | | | | | | | | X |
| **Biological sampling: Fasting blood samples[d]** | | | | | | | | | | | X | | | | | | X | | | | | | | | | | (X) |
| **Tests of physical fitness and motor functions** | | | | | | | | | | | | | | | | | | | | | | | | | | | |
| Yo-Yo Intermittent Recovery Level 1 Children's test (Yo-Yo IR1C) | | | | | | | | | | | X | | | | | | X | | | | | | | | | | X |
| Handgrip strength | | | | | | | | | | | X | | | | | | X | | | | | | | | | | X |
| 20-m sprint test | | | | | | | | | | | X | | | | | | X | | | | | | | | | | X |
| Vertical jump height - counter-movement jump | | | | | | | | | | | X | | | | | | | | | | | | | | | | X |
| Balance - postural sway | | | | | | | | | | | X | | | | | | | | | | | | | | | | X |
| Test of lower extremity gross motor function | | | | | | | | | | | X | | | | | | | | | | | | | | | | X |
| Test of upper extremity gross motor function | | | | | | | | | | | X | | | | | | | | | | | | | | | | X |
| Canadian Agility and Movement Skill Assessment (CAMSA) test | | | | | | | | | | | X | | | | | | | | | | | | | | | | X |
| **Tests of cognitive functions: Executive function, memory, attention** | | | | | | | | | | | X | | | | | | | | | | | | | | | | X |
| **Accelerometry: 24 hour/7-day thigh-positioned accelerometer** | | | | | | | | | | | X | | | | | | | | | | | | | | | | X |
| **Child questionnaires** | | | | | | | | | | | | | | | | | | | | | | | | | | | |
| Wellbeing and quality of life (KIDSCREEN-27), leisure time sport | | | | | | | | | | | X | | | | | | X | | | | | | | | | | X |
| Physical literacy (MyPL) | | | | | | | | | | | X | | | | | | | | | | | | | | | | X |
| Food literacy, school meal culture, family meal culture | | | | | | | | | | | X | | | | | | X | | | | | | | | | | X |
| **Parental questionnaires and dietary record** | | | | | | | | | | | | | | | | | | | | | | | | | | | |
| Family sociodemographic background | | | | | | | | | | | X | | | | | | | | | | | | | | | | |
| Child leisure time sports participation | | | | | | | | | | | X | | | | | | X | | | | | | | | | | X |
| Strengths and Difficulties questionnaire | | | | | | | | | | | X | | | | | | X | | | | | | | | | | X |
| Children's Sleep Habits Questionnaire | | | | | | | | | | | X | | | | | | X | | | | | | | | | | X |
| Modified SCREENS questionnaire | | | | | | | | | | | X | | | | | | X | | | | | | | | | | X |
| Child dietary habits | | | | | | | | | | | X | | | | | | X | | | | | | | | | | X |
| 3-day dietary record | | | | | | | | | | | X | | | | | | X | | | | | | | | | | X |
| **App-based measurement of screen-time (14 days)** | | | | | | | | | | | X | | | | | | X | | | | | | | | | | X |
| **School performance: Math and reading comprehension** | | | | | | | | | | | X | | | | | | | | | | | | | | | | X |
| **School absence[e]** | | | | | | | | | | | ◆━━━━━━━━━━━━━━━━━━━━━━━━━━◆ | | | | | | | | | | | | | | | |

[a]Randomization was performed at two time points to ensure timely allocation of schools to their respective groups. *First randomization*: 10 schools. *Second randomization*: 14 schools [b]Participant recruitment occurs throughout the study period, as the study is an open cohort where new pupils entering the included classes are invited for the study, and existing pupils who did not enroll at baseline can enroll later if they change their mind. [c]All data collection procedures are described in detail in Additional File 2. [d]Blood samples will only be obtained from children in Capital and Zealand Regions, i.e. in 12 schools. At 2nd follow-up assessment (T2), blood sampling will only be obtained if logistics and funding allow. [e]Absence recordings will be obtained throughout the intervention period.

**Fig 1. SPIRIT schedule of enrolment, interventions and assessments in Generation Healthy Kids.**

Based on two previous Danish school-based studies from our research groups [61, 62], we assumed a cluster size coefficient of variation of 0.24, a standard deviation of 4.65 kg for FM in both groups [61], and an intra-cluster correlation coefficient of 0.03 for FM [62]. We also assumed a correlation between FM at baseline and follow-up of 0.8. Finally, we additionally controlled for type-1 error by lowering alpha to 0.025 in the power calculation, due to a

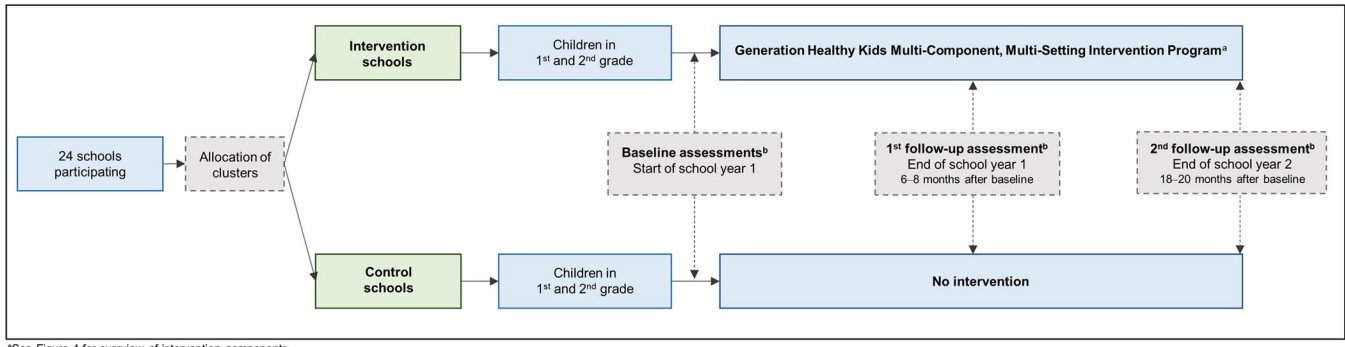

[a]See Figure 4 for overview of intervention components.
[b]See Figure 1 for full SPIRIT schedule of enrollment, interventions and assessments.

**Fig 2. Overview of study design of Generation Healthy Kids.**

planned correction of small (cluster) sample using the Satterthwaite correction for mixed linear models [63].

Under these assumptions, the study has a power of 80% to detect a significant mean difference of -0.81 kg FM or greater at 18–20 months between intervention and control. A prior Danish study found that an increase in FM of 1 kg at age 10 years was associated with a 12–15% increased risk of type 2 diabetes at age 50 [64]. Thus, our study is powered to detect a small, but clinically relevant, between-group difference in FM at 18–20 months after baseline. Statistically, using FM at follow-up as the outcome while adjusting for FM at baseline is equivalent to using the change in FM as outcome once the baseline FM has been adjusted for.

## Invitation of schools (study sites)

Schools for the study were invited from three out of five regions in Denmark: The Capital, Zealand and Southern Denmark regions. For logistic reasons, it was à priori decided to include 12 schools from Eastern Denmark (i.e. Capital/Zealand Region) and 12 schools from Western Denmark (i.e. Region of Southern Denmark).

We obtained information from the Ministry of Children and Education on all public and private schools in Denmark, including information on the number of children per year, and the proportion of children in each year whose parents had vocational education as their highest completed education [65]. We then used a snowballing method by which additional schools were invited until the target of 24 schools had been reached. Invitations were sent over three rounds in which the study area and invitation criteria were gradually expanded. Fig 3 provides a flowchart of the school invitation process, and S1 Table in S2 File shows the municipalities from which schools were invited.

In *school invitation round 1*, we invited public schools from 15 municipalities in the Capital and Southern Denmark Regions. We only invited schools with ≥2 classes per year group and schools where ≥20% of pupils in the relevant classes had parents with basic educational level, because we wanted to target schools with a high proportion of socioeconomically disadvantaged children. In *school invitation round 2*, we invited public schools from 11 additional municipalities in the Capital and Southern Denmark Regions with no socioeconomic restrictions on parental educational level. Finally, in *school invitation round 3*, we invited all schools from remaining municipalities in the Capital and Southern Denmark Regions, except three municipalities with existing school health projects and five municipalities located geographically far from the study centers. We additionally included nine municipalities in the Zealand Region with the closest geographical proximity to the study centers, and in this invitation round, we also invited private schools and schools with only one class per year group. A total of 496 schools were invited (Fig 3; S1 Table in S2 File).

Schools were contacted directly by e-mail to the school principal and/or head of primary school. If a research team member had a contact in the municipality, we contacted the municipality who then invited schools on our behalf. Schools who did not respond to our invitation received a reminder after 1–2 weeks, and if they still did not respond, we attempted to reach them by telephone. The study was also advertised in social media posts (LinkedIn), and research staff actively used their networks to generate interest for the study. When a school expressed interest in participating, a meeting was held between research team members, the school principal, and in some cases teacher representatives. Aims and content of the study were discussed, including requirements for the school to participate (see list of requirements for school participation in S2 Table in S2 File). Following this dialogue, schools were included for randomization if they decided to participate.

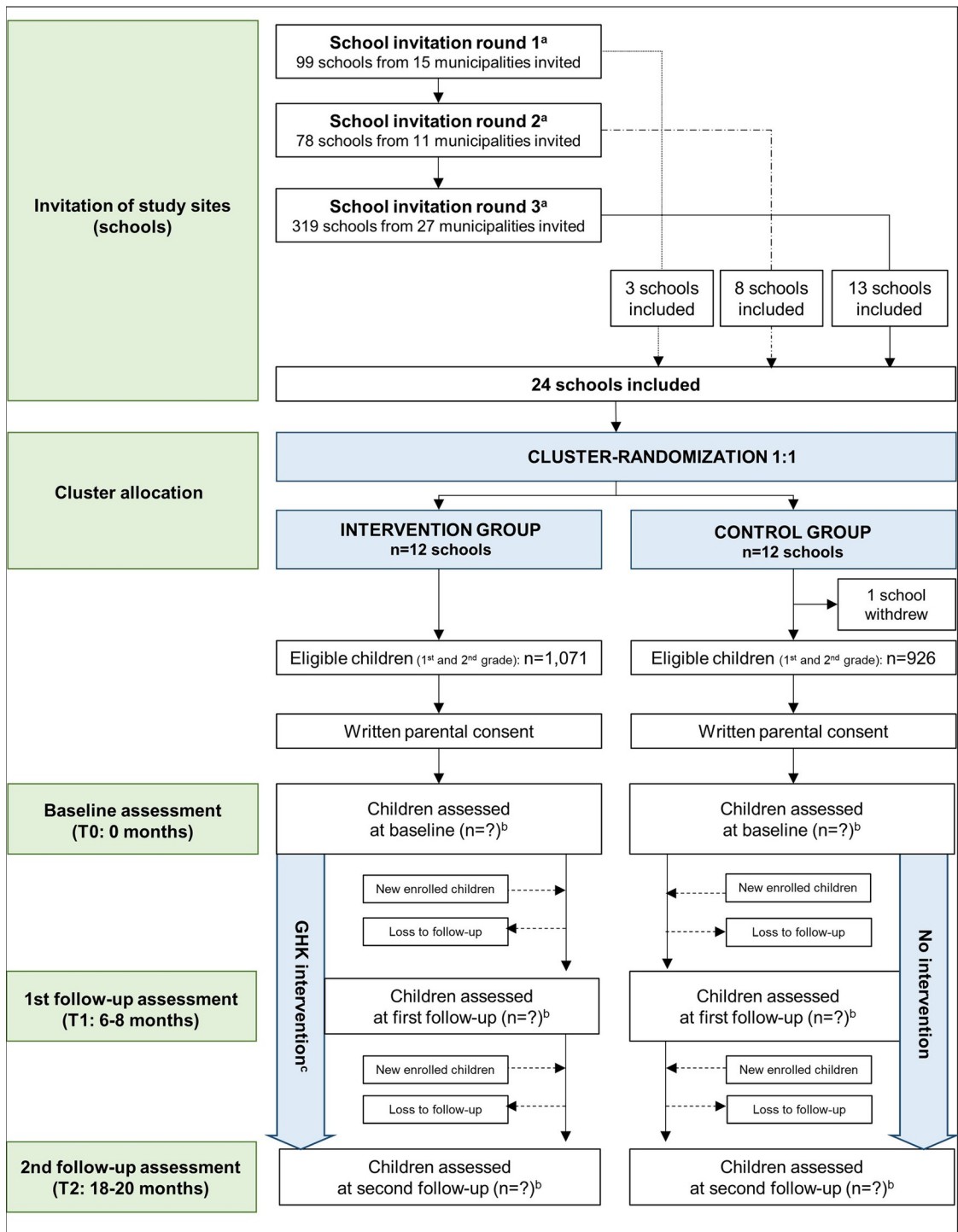

<sup></sup>ᵃCriteria in invitation round 1: Only public schools; ≥40 pupils per year group; ≥20% of parents with only vocational education. Criteria in invitation round 2: Only public schools; ≥40 pupils per year group; no socioeconomic restriction. Criteria in invitation round 3: Public and private schools (international schools and schools for children with special needs excluded); ≥20 pupils per year group; no socioeconomic restriction.
ᵇSee Figure 1 for full list of assessments in each assessment round.
ᶜSee Figure 4 for overview of contents of the Generation Healthy Kids (GHK) intervention.

**Fig 3. Flowchart of schools and participants in Generation Healthy Kids.**

## Randomization and blinding

Randomization occurred at school level prior to inclusion of individual participants (i.e., children). The 24 schools were randomly assigned to intervention or control group in a 1:1 ratio (Fig 3). A covariate-constrained randomization procedure was carried out to maximize school level covariate balance between intervention and control group and to increase the probability that child baseline characteristics did not differ substantially between groups.

Allocation was conducted in two strata (1: West, i.e. Region of Southern Denmark and 2: East, i.e. Capital/Zealand Region) and was constrained by the following school-level covariates: Number of children in 1st and 2nd grade, proportion of children with parents of basic educational level (vocational education or lower), proportion of children with non-Danish background, rural or urban municipality, and number of planned weekly physical education lessons. The school-level covariates were obtained from the Ministry of Children and Education [65], and information on physical education lessons was provided by school principals.

To ensure timely allocation of schools to their respective groups and allow schools for sufficient time to plan the subsequent school year, randomization was conducted in two steps, with the first step based on the first 10 schools recruited, and the second step based on the last 14 schools recruited. Randomization was performed using the 'cvcrand' command in STATA for covariate-constrained randomization in cluster-randomized trials [66]. Randomization was carried out by a statistician not otherwise involved in the study, and the procedure was concealed from the investigators.

Due to the nature of the intervention, the study is open label, i.e. allocation is not blinded to participants, school staff, local communities, or research team members.

After randomization, one control school decided to withdraw from the study, leaving 12 intervention and 11 control schools in the study (Fig 3).

## Characteristics of schools in the intervention and control group

Table 1 shows selected characteristics of the 23 participating schools. The number of included classes per school (1st and 2nd grade) ranges from 2–6 in the intervention group and from 2–7 in the control group. At baseline, a total of 1,071 and 926 children are eligible for the study in the intervention and control groups, respectively. The mean proportion of parents with a basic educational level is similar at intervention and control schools (intervention: 36%; control: 30%), and the mean proportion of parents with non-Danish origin is also similar in the two groups (intervention: 14%; control: 11%). Finally, both in the intervention group and control group, the majority of schools are located in urban areas.

## Study participants

**Inclusion and exclusion criteria.** All children attending 1st or 2nd grade in the participating schools are eligible for the study, and no exclusion criteria will be applied. If parents, teachers, or the study's clinically responsible physician (CM) judge that a child cannot participate in certain parts of the intervention program or measurement schedule, e.g. due to severe allergies, chronic diseases or mental/physical disabilities, the child will be eligible for the remaining parts of the study. New pupils enrolled in the participating classes during the study will be invited to participate, and if the custody holders provide consent, the child will be included in the next measurement round. In addition, eligible children whose parents decide not to enroll at baseline can enroll into the study at any later time during the study, if they change their mind about participation (Fig 3).

**Recruitment and enrolment of children.** Parents of children in the included classes will receive invitations and written participant information about the study through the online

**Table 1. Characteristics of schools in intervention and control group.**

| Group | Region[a] | Municipality | Rando-mization round | Public or private school? | Number of classes included | Number of eligible children at baseline | % children with parents having basic education[b,c] | % children with non-Danish origin[b] | Urban/rural location | Expanded local community intervention? |
|---|---|---|---|---|---|---|---|---|---|---|
| Intervention | East | Rødovre | 1 | Public | 6 | 134 | 44% | 33% | Urban | Yes |
| | East | Høje-Taastrup | 1 | Private | 6 | 134 | 21% | 10% | Urban | Yes |
| | East | Frederiksberg | 2 | Private | 2 | 44 | 0% | 0% | Urban | Yes |
| | East | Frederiksberg | 2 | Public | 6 | 125 | 20% | 23% | Urban | Yes |
| | East | Glostrup | 2 | Public | 4 | 54 | 39% | 23% | Urban | Yes |
| | East | Slagelse | 2 | Public | 2 | 52 | 52% | 0% | Urban | No |
| | West | Nordfyns | 1 | Public | 4 | 97 | 65% | 6% | Rural | Yes |
| | West | Kerteminde | 1 | Public | 4 | 78 | 51% | 18% | Rural | Yes |
| | West | Kerteminde | 2 | Public | 2 | 43 | 34% | 0% | Rural | Yes |
| | West | Odense | 1 | Public | 4 | 99 | 26% | 10% | Urban | No |
| | West | Odense | 2 | Public | 5 | 121 | 22% | 16% | Urban | No |
| | West | Varde | 2 | Public | 4 | 90 | 55% | 25% | Rural | No |
| Control | East | Hillerød | 1 | Public | 4 | 93 | 33% | 20% | Urban | *Not applicable* |
| | East | Lyngby-Tårbæk | 2 | Public | 4 | 57 | 26% | 27% | Urban | *Not applicable* |
| | East | Hørsholm | 2 | Public | 4 | 84 | 8% | 0% | Urban | *Not applicable* |
| | East | Slagelse | 2 | Public | 3 | 46 | 31% | 0% | Urban | *Not applicable* |
| | East | Lejre | 2 | Public | 7 | 142 | 17% | 7% | Urban | *Not applicable* |
| | West | Odense | 1 | Public | 4 | 94 | 33% | 8% | Urban | *Not applicable* |
| | West | Odense | 1 | Public | 6 | 143 | 25% | 10% | Urban | *Not applicable* |
| | West | Varde | 1 | Public | 2 | 37 | 37% | 0% | Rural | *Not applicable* |
| | West | Faaborg-Midtfyn | 1 | Public | 4 | 69 | 49% | 18% | Rural | *Not applicable* |
| | West | Faaborg-Midtfyn | 2 | Public | 4 | 69 | 23% | 9% | Rural | *Not applicable* |
| | West | Vejen | 2 | Public | 4 | 92 | 52% | 22% | Rural | *Not applicable* |

[a]East encompasses Capital and Zealand Regions. West encompasses the Region of Southern Denmark.

[b]Average for 1st and 2nd grade in school year 2020/21, i.e. latest data available at time of school recruitment and randomization.

[c]Basic education is defined as vocational education or lower.

communication platform used for parent-school communication in Danish schools ("Aula/ Intra"). Invitations will also be handed out in printed form from teachers. Information meetings about the study for parents and children will be held at all participating schools. At the meetings, parents will receive oral information about the intervention and planned study procedures, and information will be provided to children in a language and at a level appropriate for their age and comprehension. Families who are unable to participate in the information meetings will be offered the chance to participate in online meetings or to have the information in person. Furthermore, research staff will visit after-school clubs during the time where parents usually pick up their children to provide participant information to parents and children who were not able to attend the information meetings, and to answer questions about the study. Finally, research team members will also visit school classes during school hours to explain the study intervention and procedures to the children.

Written informed consent will be obtained from all custody holders before enrolment of children into the study. In compliance with the Danish Law on the Health Research Ethics,

informed consent will only be deemed valid if the custody holders have received oral and written participant information, *and* the child has been informed orally about the study in a language suitable for children. Prior to provision of informed consent, parents will be asked to discuss the study with their child and take the child's perspectives on participation into account. Custody holders and children will be informed that participation is entirely voluntary, and that consent can be withdrawn at any time during the study. For children with two custody holders, both are required to provide informed consent, or alternatively one custody holder can grant a written power of attorney for the other to consent on their behalf. The informed consent will be obtained by trained research staff with in-depth knowledge of the intervention and study procedures.

**Withdrawal criteria.** Parents can withdraw their child from the study at any time during the study period. In this case, the child will be discontinued from the measurement schedule and no further data will be collected, but children in the intervention group will still be offered the intervention activities implemented as part of the normal school curriculum. Data collected before withdrawal will be included in the study.

## Intervention

**Overall intervention structure.** Fig 4 provides a schematic illustration of the structure and components of the GHK intervention, and S3 Table in S2 File) provides a detailed description of the settings, target groups, planned timing and frequency of all intervention components. The intervention involves multiple settings, including families, schools, after-school clubs, and local communities around the schools. In each setting, the intervention addresses the focus areas of diet, physical activity, screen media use and sleep habits. All intervention components are centered around the overarching goal of promoting child healthy weight and wellbeing.

As shown in Fig 4 and S3 Table in S2 File), some intervention components are predefined based on existing evidence and the GHK pilot study, and these components will be implemented in a uniform and standardized manner at all intervention schools ("core elements"). Other intervention components will be co-created in a participatory process with each intervention school and community and will thus differ between schools and communities based on local needs and relevance ("co-created elements"). In accordance with the Supersetting approach [55], the specific content of the co-created elements will be decided through direct interaction and dialogue with local stakeholders, including school management, school staff, parents, and local community members, to ensure that the intervention components are sensitive to the local context and take into account what is considered relevant and feasible locally. We will furthermore use principles from the Supersetting approach to integrate and coordinate the various intervention components in different settings. In practice, this will be done by implementing common themes such as "Family and community", "Nature and outdoor life", "Play and children's perspectives". The themes will run for 2–3 months each during the intervention period and will be applied across all settings in each intervention community to promote integration between project activities (Fig 4), as inspired by a previous Danish community-based health promotion project [54].

To ensure equal treatment and prevent any children from feeling excluded or left out, the intervention will be made available to all children in the involved classes regardless of their participation in the study's biomedical measurement schedule. Thus, all children will be offered the free school lunch and extra physical activity during school hours as detailed in the following sections. Likewise, all families in the involved classes will be invited for family events

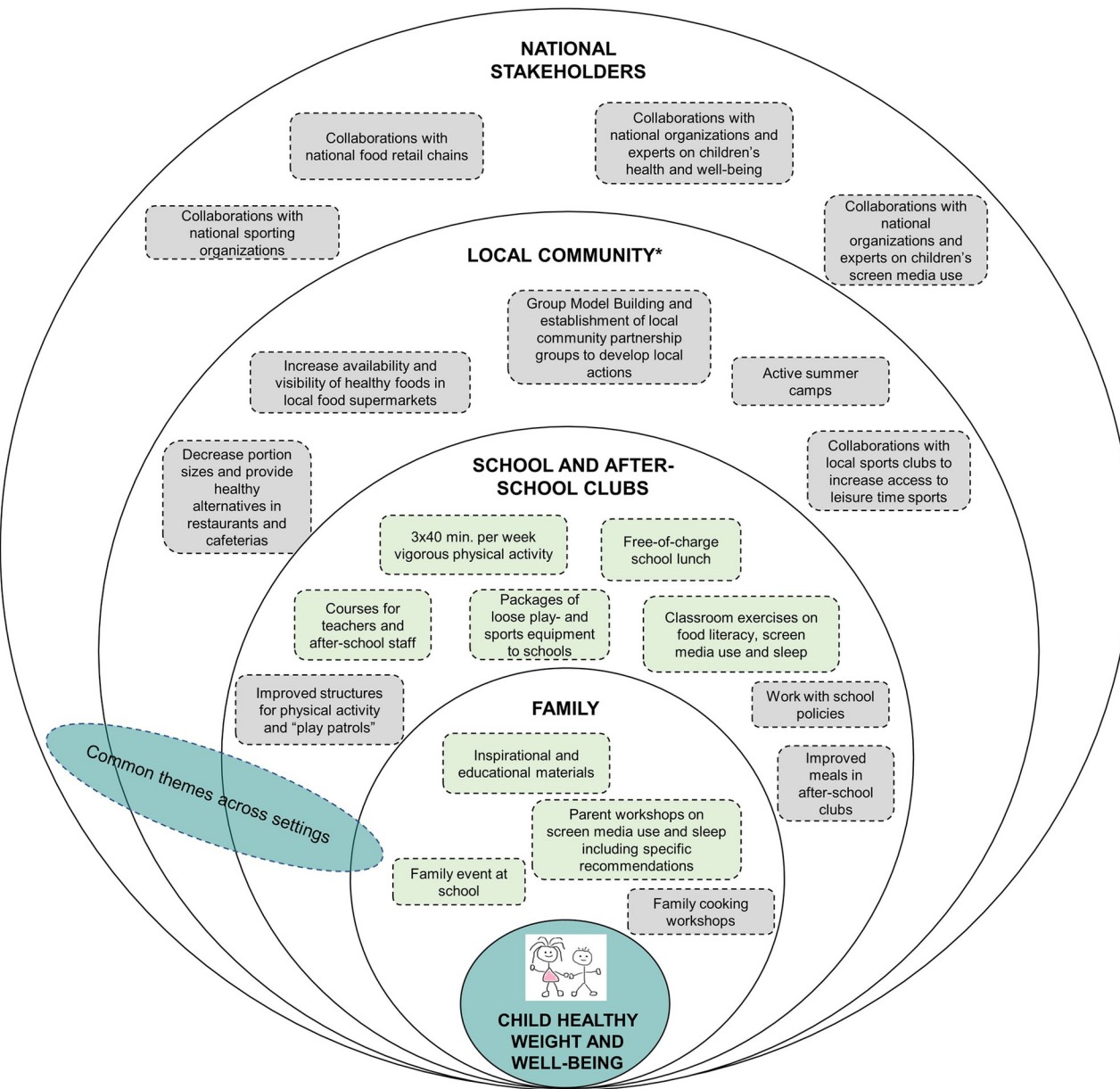

Note: Intervention components marked with green are core components implemented uniformly at all intervention schools. Intervention components marked with gray are developed in co-creation with local and national stakeholders.
*The local community intervention will focus on the local community around eight of the 12 intervention schools.

**Fig 4. Overview of structure and components of the GHK intervention.**

organized as part of the study, irrespective of whether their child is enrolled in the measurement protocol.

The following sections describe the intervention components in each setting in greater detail.

**Interventions in schools and after-school clubs.** *Diet.* A core component of the dietary intervention is a free-of-charge school lunch which will be served in all participating school classes four days per week during the intervention period (Fig 4; S3 Table in S2 File). Meals

will be based on the Danish climate-friendly food-based dietary guidelines and the Nordic Nutrition Recommendations and will be rich in fruit, vegetables, wholegrains and fish, and low in sugar and saturated fat. Furthermore, focus will be on drinking water [39, 40]. The lunch will be mainly bread-based, which is the typical lunch in Denmark, and will take a limited food budget into account. In order to improve food courage and liking of healthy foods, the meals will be developed specifically for the age group and will be designed to combine familiar tastes with new and potentially unfamiliar food items. Children will eat in smaller groups as families and will be actively involved in table setting, assembling the meal, and cleaning up afterwards–a meal concept that was developed and tested during the GHK pilot study. Schools are asked to allocate approximately 45 minutes for the lunch break, and teachers will be educated to act as meal hosts to support a pleasant and educational eating situation.

Teachers will receive food literacy exercises for use in the classroom to support children's knowledge and practical skills in relation to food, tastes and cooking [67]. These materials were developed in the GHK pilot and include e.g. sensory taste exercises and vegetable rhymes. Further elements in the school dietary intervention include ensuring access to cold drinking water and, if relevant in the local setting, working with after-school clubs to improve the nutritional quality of their offers of breakfast and afternoon snacks. This intervention component will be tailored to each after-school club depending on local needs, relevance, and resources, based on a close dialogue with afterschool club management and staff. Where feasible, other activities focusing on healthy foods will be arranged in collaboration with after-school clubs, e.g., excursions to local greengrocers or supermarkets and/or afternoon events for families (Fig 4; S3 Table in S2 File).

*Physical activity.* A core intervention element is three weekly 40-minute sessions of organized vigorous physical activity implemented during school hours. The sessions will encompass modified and varied sporting activities using the FIT FIRST 10 concept [68], which is based on prior studies [60, 69] and was tested in the GHK pilot. The sessions are based on adaptations of ten sports commonly played in Denmark (e.g., football, team handball, badminton, judo), and the activities are developed to promote motivation and active involvement of all children regardless of fitness levels and prior sporting experience. Sessions will be delivered by schoolteachers after thorough training by research staff (Fig 4; S3 Table in S2 File).

A further core component of the intervention is to improve children's opportunities for physical activity during school recess. To achieve this, we will distribute packages of loose play and sports equipment to all schools. Furthermore, based on a local assessment of needs and relevance we will work with schools and after-school clubs to improve structures for active outdoor play, e.g. by drawing lines for ball games in school yards and/or providing small goals for football, floorball (unihockey), team handball etc. [70]. If feasible and relevant locally, older pupils at the intervention schools will be educated to organize and supervise physically active games during recess with children in the participating classes (so-called "play patrols") (Fig 4; S3 Table in S2 File).

*Screen media use and sleep practices.* During the intervention period, teachers will conduct three instructional courses on screen media use and sleep practices with children in the classroom. Each instructional course will comprise three lessons. The first course addresses the importance of sleep and sleep hygiene. The second course highlights family screen habits, encompassing individual screen usage, screen regulation within the household, and role modeling behaviors. The final course delves into online communities and communication, as well as the functioning of algorithms (Fig 4; S3 Table in S2 File). The classroom exercises will encourage children to reflect upon screen media and sleep practices in their family, and the results will be presented at later workshops for parents (see below, "Interventions focusing on families").

*Courses for teachers and after-school staff.* To promote motivation and ownership and prepare school staff for implementing the intervention activities, teachers and other school staff

will participate in three face-to-face courses at the beginning of the intervention period. The courses focus on:

1. *Diet and nutrition*, including the official Danish nutrition guidelines, food pedagogics, principles and practical aspects of the school lunch program, and inspiration for working with healthy meals in after-school clubs (four hours);

2. *Physical activity*, including methods to promote children's physical literacy and practical implementation of FIT FIRST 10 (six hours);

3. *Screen media use and sleep habits*, focusing on evidence-based recommendations for children's sleep and screen media use, as well as examples of classroom exercises on digital literacy (three hours).

All courses were developed and evaluated during the GHK pilot study. Brush-up courses will be offered at the start of school year two for new staff, either online or as face-to-face courses (Fig 4, S3 Table in S2 File).

*School health policies*. Existing school policies focusing on children's health and wellbeing will be mapped by the research team during intervention year one. In the subsequent year, research team members will provide schools with guidance on developing and implementing supplementary actions within the focus areas of GHK. Possible examples include approaches focusing on limiting screen media use during school time and in after-school clubs, increasing active outdoor play, and promoting healthy school food environments. The focus of this development process will be on ensuring sustainment and further context-specific adaptation of GHK intervention components after the study concludes (Fig 4, S3 Table in S2 File).

**Interventions focusing on families.** *Inspirational and educational material.* During the intervention period, parents will receive written inspirational and educational materials within the intervention's focus areas. The materials were developed during the GHK pilot study and include evidence and recommendations about children's screen media use and sleep practices; suggestions for activities to support dialogue about screen media and sleep practices in the family; and simple recipes and inspiration to support healthy eating and snacking habits in the family. Materials will be sent electronically to parents and distributed in printed form by teachers (Fig 4, S3 Table in S2 File).

*Family workshops and events.* Several workshops and events for parents and/or families will be arranged at the school (Fig 4; S3 Table in S2 File). These include:

- A family event covering all four focus areas of GHK during the first school year, including e.g. tips and tricks for healthy snacking habits in the family, ideas for fun physically active games for parents and children, and dialogue exercises on family screen media use.

- Three parent workshops on children's screen media use and sleep practices. The parental workshops will focus on providing specific evidence-based recommendations for screen media and sleep habits, informing parents about digital literacy and their child's screen media habits, and establishing a dialogue on the subject among parents. The Danish non-government organization Just Human [71] will develop and deliver the parent workshops in collaboration with the research team. The teaching material builds on existing material from several Danish organizations in the field and was tested during the GHK pilot.

- If feasible and relevant locally, family cooking workshops may be organized in which parents and children are invited to cook together following simple and hands-on healthy recipes to gain practical healthy food competences in the families (Fig 4, S3 Table in S2 File).

**Interventions in the local community.** *Collaborations with local sports organizations and active summer camps.* At all intervention schools, we will promote collaborations between schools, after-school clubs, and local leisure-time sports clubs and organizations. The focus will be on supporting children who are not currently members of a sports club to join. Depending on the local context, this may involve advertising existing "Free-pass"-schemes (which allow children from socioeconomically disadvantaged families to participate in leisure time sports free of charge) and/or organizing visits from sports organizations in the after-school club. Furthermore, in collaboration with sports organizations and after-school clubs, summer camps incorporating physical activity will be organized during the summer holiday where children are introduced to various games and sports. The emphasis will be on activities that are inclusive and adapted to children with limited sporting experience (Fig 4; S3 Table in S2 File).

*Expanded local community intervention: Building community capacity and establishing Local Community Partnership Groups.* At eight of the 12 intervention schools (located in six different municipalities), we will conduct an expanded local community intervention involving multiple stakeholders from the local community around the school. For the remaining four intervention schools, a control school is located in the same municipality (Table 1). To avoid contamination between clusters, these schools will not receive the expanded community intervention.

The local community intervention is based on systems thinking and inspired by the WHO STOPS trial [26]. The overall goal is to build local capacity to initiate and sustain actions to improve child health and wellbeing in the community. In each community we will initially perform a stakeholder analysis to identify community representatives whom it would be relevant to engage in the intervention. Through dialogue with the school and municipality, we will map potentially relevant local stakeholders, including representatives from the municipality, school leader, members of the school board, parent representatives from the participating classes, leisure time organizations, sports clubs, social housing organizations, other non-governmental organizations, food retailers, restaurants, and local media. To raise local awareness and to engage and recruit community members, we will furthermore collect and communicate local data on child health and wellbeing, and a local evidence brief will be prepared for each community.

All identified stakeholders will be invited to participate in two workshops. In the first workshop, community members will together build a causal loop diagram of the drivers of childhood overweight and obesity in their local community using Group Model Building [57] and STICKE software [58]. In the second workshop, community members will collaborate to design and prioritize actions to promote children's healthy weight and wellbeing in the local community, inspired by the systems map generated in the first workshop.

After the workshops, a Local Community Partnership Group (LCPG) will be established consisting of selected local stakeholders and leaders in each community. The LCPGs will be responsible for developing and implementing activities within the local community that can promote healthier environments for children and families, including healthier food environment and easier access to active living. This will be done across multiple settings within the local communities. Activities within each setting will be coordinated and integrated with activities in other settings with the aim to achieve synergistic effects, using the Supersetting approach [55]. The LCPG will function as a steering group during the intervention period and will help ensure coordination and long-term local anchoring of the actions initiated in each community (Fig 4; S3-S5 Tables in S2 File).

*Involvement of children.* We will seek to actively include children's own perspectives on how their school and local community can enhance their health and wellbeing. During school

hours, research team members will facilitate class discussions with selected classes regarding the children's own perceptions of health and wellbeing, including the children's viewpoints on what constitutes a "healthy" life as well as barriers and facilitators for thriving and having a healthy life. The process will be guided by participatory action research methods, and we will use the method "Future Workshops" where children are encouraged to establish their vision for a healthier and better school and local environment [72]. We will then hand over the children's visions to the school management, school board and the LCPGs, and the research group will work to facilitate and support the LCPGs in implementing sustainable local changes based on the children's ideas and wishes, where feasible and relevant (Fig 4; S3 Table in S2 File).

*Intervention to promote a healthy food environment*. The goal of this intervention component is to promote healthy food environments which support families in healthy eating by making healthy foods more available, affordable, and accessible in the local community [73]. In particular, we will focus on increasing availability and sales of fruits, vegetables, wholegrains and fish, and decreasing sales of sweets and sugar-sweetened beverages.

Initially, we will perform a mapping of local supermarkets, restaurants and fast-food outlets in each intervention community using a Danish administrative database of food retailers [74]. Identified stakeholders from the food environment will be invited to join the workshops and Local Community Partnership Groups (see above, section "Building community capacity and establishing Local Community Partnership Groups"). In collaboration with local supermarkets and restaurants, we will develop and implement structural initiatives to increase sales of healthy foods for selected time periods during the intervention period, based on our experience from a prior Danish local community project [75–77]. The initiatives will focus on nudging, pricing, and product placement in supermarkets, and on decreasing portion sizes and offering healthy alternatives in local restaurants, fast-food outlets and/or cafeterias. In collaboration with local food retailers and the LCPGs, we will furthermore seek to organize activities and events in the local community which support the overall project themes and contribute to increased local visibility and engagement of local actors in GHK [54].

## Control group

At the control schools, the school day will continue as normally, and no intervention will take place. To promote continued motivation for control schools to participate in the study, all control schools will be offered courses for teachers of the oldest pupils (7th-9th grade) in FIT FIRST sessions tailored for this age group (FIT FIRST teen). Furthermore, all children in the participating classes will be offered to participate in an activity day with fun physical activities, arranged by the research team after the end-of-study measurements. Finally, all children participating in the biomedical measurement schedule will receive a small gift after each measurement round.

## Data collection

We will collect comprehensive quantitative and qualitative data from various data sources to investigate the intervention's effect and to evaluate implementation of the intervention.

**Effectiveness data.** To evaluate the effectiveness of the intervention, measurements and data will be collected at three time points: At baseline before initiation of the intervention (*baseline assessment)*, after the first school year (*first follow-up assessment*: 6–8 months after baseline), and after the second school year (*second follow-up assessment*: 18–20 months after baseline) (Figs 1–3). Some measurements will be performed at all time points, while others will only be performed at selected assessment rounds. Effectiveness data will be collected at schools

during school time and at home. Fig 1 provides an overview of assessments at each time point, and S2 File gives a detailed description of measurement procedures.

*Effectiveness data collected from children during school time*. Measurements during school time will be spread over 2–4 days per assessment round for each child. The measurements will be performed by trained test teams at the school and/or nearby sports facilities using standard operating procedures. All staff in contact with children will undergo pedagogical and professional training, and measurements will be undertaken in a safe and pleasant atmosphere. On measurement days, if a child does not want to participate or expresses discomfort or uneasiness with the procedure, the measurement will not be performed, even though there is parental consent.

*Body composition, anthropometry, blood pressure and resting heart rate*. Body composition will be measured using air displacement plethysmography (BODPOD GS-X, COSMED Srl., Rome, Italy) and a bioimpedance analyzer (InBody270, InBody, California, USA). The bioimpedance instrument will also be used to measure body weight. Height will be measured using a portable stadiometer, and waist circumference using a measuring tape at the level of the umbilicus. Finally, we will measure resting blood pressure and heart rate in the supine position after a 10-minute rest (Fig 1; S2 File).

*Physical fitness and gross motor functions*. Aerobic fitness and intermittent exercise performance will be evaluated by the Yo-Yo Intermittent Recovery Level 1 Children's test [78]. Muscular fitness and strength will be evaluated using a handgrip dynamometer, and sprint performance will be determined using a 20-m sprint test. Balance ability will be evaluated as postural sway during maintained balance, and lower-extremity gross motor function will be evaluated as maximum reaching ability during unilateral maintained balance. Furthermore, we will measure jump height using the vertical counter-movement jump test. Upper-extremity gross motor function will be measured using a test of speed and accuracy in the ability to perform accurate, goal-directed reaching movements towards visually displayed targets. Finally, agility will be assessed using the Canadian Exercise and Movement Skills Ability test (CAMSA) [79] (Fig 1; S2 File).

*Wellbeing, physical- and food literacy, cognitive function and school performance*. Children will be asked to complete a video-assisted questionnaire containing questions on self-reported wellbeing and quality of life (KIDSCREEN-27) [80], physical literacy (the MyPL questionnaire) [81], leisure time sports participation, and a questionnaire on food literacy and meal culture developed specifically for GHK. Furthermore, to assess cognitive function, children will be asked to complete a battery of standardized, validated and age-appropriate neurocognitive tests of executive function, memory and attention. To assess school performance, standardized and age-appropriate tests of mathematics and reading comprehension will be administered by schoolteachers on a separate day during school time (Fig 1; S2 File).

*Blood samples*. All children participating in the Capital and Zealand Regions (six intervention schools, five control schools) will be offered to provide a blood sample, if their legal guardians give informed consent for this procedure. Venous blood samples (max. 30 ml) will be drawn from the child's forearm in the morning in the overnight fasting state. Sampling will be performed by trained staff with experience with blood sampling in young children, and local anesthetic patches will be applied prior to sample collection. Blood samples will be stored at maximum -70°C and analyzed for markers of intake and nutrient status, growth and development, cardiometabolic health, inflammation, cognition, as well as genetics and epigenetics as further described in S2 File. A research biobank will be established for this study until analysis, located at the University of Copenhagen. Furthermore, a biobank will be established for future research purposes, containing only samples from children whose parents provide separate informed consent.

*School recordings of child absence*. The schools' recordings of child absence during the intervention period will be obtained with parental consent as a proxy for school-related wellbeing, school motivation and general child health [82].

*Effectiveness data collected at home*: *Parental questionnaire*. Parents will be asked to complete an electronic questionnaire with items on family sociodemographic background, child's leisure time sports participation, child's dietary habits, child's wellbeing, child and parent screen media practices, and child's sleep habits. Child wellbeing will be assessed using the Strengths and Difficulties questionnaire [83], sleep habits will be assessed with the Children's Sleep Habits Questionnaire [84], and family screen media practices will be assessed using a modified version of the SCREENs questionnaire [48] (Fig 1; S2 File).

*Three-day dietary record*. Parents will be asked to complete a 3-day dietary recording of their child's intake of food and beverages during two weekdays and one weekend day using a validated, web-based tool (myfood24®, Dietary Assessment Ltd., Leeds, United Kingdom) (Fig 1; S2 File).

*Accelerometry*. To objectively assess daily physical activity, sedentary time and sleep habits, children will be asked to wear a thigh-positioned accelerometer (Axivity AX3, Axivity Ltd., United Kingdom) for 24 hours during seven consecutive days. Details of the accelerometry procedure are found in S2 File.

*App-based measurement of screen time*. At baseline, parents will be asked to install an app (Ethica©, Toronto, Canada) on their child's smartphone and/or tablet (if the child has one). The Ethica app is developed for research purposes and is able to measure screen time on a second-to-second basis. At all three measurement rounds, we will collect data on children's screen media pattern over a period of 14 consecutive days (Fig 1; S2 File).

*Effectiveness data from supermarkets*. The effectiveness of the supermarket intervention will be evaluated in a quasi-experimental design using supermarket sales data. For each supermarket in GHK intervention communities that agrees to participate, we will select a control supermarket from the same retail chain located in a neighboring municipality with similar population density and socioeconomic characteristics, and with similar annual turnover as the intervention supermarket. This quasi-experimental design is chosen for the food environment intervention because the cluster-randomization in GHK was performed at school-level, not supermarket level. Therefore, supermarkets in intervention- and control areas of GHK may not be comparable. Weekly data on sales of healthy food items targeted by the intervention, e.g., fruits and vegetables, will be obtained from intervention- and control supermarkets during the intervention period and the year before. To investigate possible substitution effects, we will obtain sales data for all overall food categories in addition to the targeted food items [75–77].

**Process evaluation data.** A thorough process evaluation will be conducted to evaluate the degree of implementation, fidelity, and acceptability of the intervention and to document implementation-related factors, organizational and wider contextual factors, and causal mechanisms which influence intervention outcomes. The process evaluation will be guided by the UK Medical Research Council Framework and guidance [56, 85], and will use key concepts from, among others, the Consolidated Framework for Implementation Research [86] and the RE-AIM framework [87]. The process evaluation will also include an assessment of the potential for sustainability and scale-up of GHK. A comprehensive description of the process evaluation protocol, including the program theory of GHK, will be published separately. In brief, process evaluation data will include field observations in schools, after-school clubs and in the local community; interviews and focus groups with parents, school staff, school management, children, and local community members; and questionnaire data from children, parents, school staff and management. To map relevant initiatives focusing on diet, physical activity,

screen media use and sleep in the control schools, school management at each control school will be asked to participate in interviews and complete questionnaires.

Implementation and stakeholder engagement in each local community will be tracked with inspiration from the methods described by Maitland et al. [88]. This will include a continuously updated action register of project activities implemented in each community, and a stakeholder engagement database including a quarterly assessment of the degree of engagement and capacity of local stakeholders involved in the project [88].

**Registry data.** In Denmark, all residents have a unique personal identification number which is assigned at birth or immigration into the country and is used universally in all public registries and administrative databases [89]. Using this personal identification number, we will obtain registry data on the study cohort from national Danish health- and socioeconomic registers under Statistics Denmark and the National Health Data Authority. The purposes of the registry linkages are: 1) to compare sociodemographic characteristics of participants with eligible non-participants to assess the generalizability of study findings; 2) to facilitate the possibility to use register data on eligible non-participants to correct for possible post-randomization selection bias; 3) to supplement survey responses on family sociodemographic characteristics and thereby limit missing data; 4) to facilitate the possibility to investigate the long-term effectiveness of the intervention on height, weight and BMI, by registry linkage with The National Child Health Register up to seven years after completion of the intervention; and 5) to study associations between lifestyle factors and biomarkers collected in the study and subsequent long-term health and morbidity among participants. All registry linkages will be performed on secure servers at Statistics Denmark or the National Health Data Authority on pseudonymized datasets.

## Outcomes

The primary outcome of the trial is the difference in change in FM (measured in kg) assessed by air-displacement plethysmography from baseline to end-of-study (after 18–20 months) in the intervention group compared with the control group (Table 2).

Several secondary and other outcomes will be evaluated, including markers of body composition, anthropometry and growth, wellbeing, cognitive functions, school performance, cardiometabolic health, tests of physical fitness and motor functions, physical activity levels and physical literacy, nutritional status, dietary intake and food literacy, screen media practices,

**Table 2. Overview of primary, secondary and other pre-specified outcomes in the Generation Healthy Kids trial.**

| Type of outcome | Category | Outcome | Description | Time frame |
|---|---|---|---|---|
| Primary outcome | Body composition | FM | Between-group difference in change in FM (kg) measured by air-displacement plethysmography (BODPOD). | Baseline, 18–20 months |
| Secondary outcomes | Body composition | FFM | Between-group difference in change in FFM (kg) measured by air-displacement plethysmography (BODPOD). | Baseline, 18–20 months |
| | | FM index | Between-group difference in change in FM index $(kg/m^2)$ measured by air-displacement plethysmography (BODPOD). | Baseline, 18–20 months |
| | | FFM index | Between-group difference in change in FFM index $(kg/m^2)$ measured by air-displacement plethysmography (BODPOD). | Baseline, 18–20 months |
| | | FFM-to-FM ratio | Between-group difference in change in FFM-to-FM ratio measured by air-displacement plethysmography (BODPOD). | Baseline, 18–20 months |
| | | % FM | Between-group difference in change in % FM measured by air-displacement plethysmography (BODPOD). | Baseline, 18–20 months |
| | | FM | Between-group difference in change in FM (kg) measured by a bioimpedance analyzer (InBody 270). | Baseline, 6–8 months, 18–20 months |
| | | FFM | Between-group difference in change in FFM (kg) measured by a bioimpedance analyzer (InBody 270). | Baseline, 6–8 months, 18–20 months |
| | | FM index | Between-group difference in change in FM index $(kg/m^2)$ measured by a bioimpedance analyzer (InBody 270). | Baseline, 6–8 months, 18–20 months |
| | | FFM index | Between-group difference in change in FFM index $(kg/m^2)$ measured by a bioimpedance analyzer (InBody 270). | Baseline, 6–8 months, 18–20 months |
| | | FFM-to-FM ratio | Between-group difference in change in FFM-to-FM ratio measured by a bioimpedance analyzer (InBody 270). | Baseline, 6–8 months, 18–20 months |
| | | % FM | Between-group difference in change in % FM measured by a bioimpedance analyzer (InBody 270). | Baseline, 6–8 months, 18–20 months |
| | Anthropometry | Height | Between-group difference in change in standing height (cm) measured using a portable stadiometer. | Baseline, 6–8 months, 18–20 months |
| | | Weight status | Between-group difference in change in prevalence of underweight, normal weight, overweight and obesity, based on cutoffs by Cole et al. [90] and the International Task Force of Obesity. Weight and height are measured by a bioimpedance analyzer (InBody 270) and a portable stadiometer, respectively. | Baseline, 6–8 months, 18–20 months |
| | | BMI z-score | Between-group difference in change in BMI z-score $(kg/m^2)$ based on WHO references. Weight and height are measured by a bioimpedance analyzer (InBody 270) and a portable stadiometer, respectively. | Baseline, 6–8 months, 18–20 months |
| | | Waist circumference | Between-group difference in change in waist circumference (mm) by non-elastic measuring tape at umbilicus level. | Baseline, 6–8 months, 18–20 months |

(*Continued*)

**Table 2.** (Continued)

| Type of outcome | Category | Outcome | Description | Time frame |
|---|---|---|---|---|
| Other pre-specified outcomes | Wellbeing and mental health | Health-related quality of life | Between-group difference in change in the total summary score of health-related quality of life, and sub-scale scores of physical well-being, psychological well-being, autonomy and parent relation, peers and social support, and school environment. Assessed by KIDSCREEN-27 (child-reported). | Baseline, 6–8 months, 18–20 months |
| | | Mental health | Between-group difference in change in the total difficulties score of the Strengths and Difficulties Questionnaire, as well as sub-scale scores of emotional symptoms, conduct problems, hyperactivity/inattention, relationship problems and prosocial behavior (parent-reported). | Baseline, 6–8 months, 18–20 months |
| | Cognitive functions | Processing speed | Performance in neuropsychological assessment of processing speed as choice reaction time, CANTAB test battery. | Baseline, 18–20 months |
| | | Sustained attention | Performance in neuropsychological assessment of sustained attention ability assessed as errors during a sustained test procedure, CANTAB test battery. | Baseline, 18–20 months |
| | Cognitive functions | Spatial working memory | Performance in neuropsychological assessment assessed as errors and strategy during the test, CANTAB test battery. | Baseline, 18–20 months |
| | | Inhibitory control | Performance in neuropsychological assessment assessed as errors during the test, CANTAB test battery. | Baseline, 18–20 months |
| | | Cognitive flexibility | Performance in neuropsychological assessment assessed as errors during the test, CANTAB test battery. | Baseline, 18–20 months |
| | | Fine motor control | Performance in neuropsychological assessment assessed as endpoint accuracy during the test, CANTAB test battery. | Baseline, 18–20 months |
| | School performance and absence | Mathematics proficiency | Between-group difference in change in mathematics proficiency, measured by Hogrefe math test. | Baseline, 18–20 months |
| | | Reading comprehension | Between-group difference in change in Danish reading skills, measured by Hogrefe reading comprehension test. | Baseline, 18–20 months |
| | | School absence due to sickness and other causes | Between-group difference in change in number of days absent from school due to sickness recorded in school registration data. | Baseline, 6–8 months, 18–20 months |
| | Cardiometabolic health | Resting blood pressure | Between-group difference in change in systolic and diastolic blood pressure (mmHg). | Baseline, 6–8 months, 18–20 months |
| | | Resting heart rate | Between-group difference in change in heart rate (beats/min). | Baseline, 6–8 months, 18–20 months |
| | | Plasma lipid profile | Between-group difference in change in plasma total, LDL and HDL cholesterol and triacylglycerol (mmol/l)[a]. | Baseline, 6–8 months, (18–20 months)[a] |
| | | Plasma insulin | Between-group difference in change in plasma insulin (pmol/l)[a]. | Baseline, 6–8 months, (18–20 months)[a] |
| | | Plasma glucose | Between-group difference in change in plasma glucose (mmol/l)[a]. | Baseline, 6–8 months, (18–20 months)[a] |
| | Physical fitness and motor functions | Aerobic fitness and intermittent exercise performance | Between-group difference in change in aerobic fitness and intermittent exercise performance as measured by Yo-Yo Intermittent Recovery Level 1 Children's test (Yo-Yo IR1C). | Baseline, 6–8 months, 18–20 months |
| | | Sprint performance | Between-group difference in change in sprint performance as measured by a 20-m sprint test | Baseline, 6–8 months, 18–20 months |
| | | Muscular fitness and strength | Between-group difference in change in muscular fitness and strength as measured by handgrip dynamometer (N). | Baseline, 6–8 months, 18–20 months |
| | | Balance ability | Between-group difference in change in balance ability as measured by sway path (m/s). | Baseline, 18–20 months |
| | | Jump height | Between-group difference in change in jump height (m) as measured by the vertical countermovement jump test. | Baseline, 18–20 months |
| | | Lower extremity gross motor performance | Between-group difference in change in lower extremity gross motor function measured as leg reaching ability during unilateral stance in a Y-balance test. | Baseline, 18–20 months |
| | | Upper extremity gross motor performance | Between-group difference in change in upper extremity gross motor function as measured by test of goal-directed reaching movements. | Baseline, 18–20 months |
| | | Agility | Between-group difference in change in agility as measured by Canadian Exercise and Movement Skills Assessment (time to completion and score). | Baseline, 18–20 months |
| | Physical activity levels and physical literacy | Physically active behavior | Between-group change in total time (minutes) spent being physically active (defined as any waking activity characterized by not being in a sitting, reclining or lying posture with minimal stationary movement), as measured by thigh-positioned accelerometer (Axivity AX3®) worn for seven consecutive days. | Baseline, 18–20 months |
| | | Moderate-to-vigorous physical activity | Between-group change in total time (minutes) spent in moderate to vigorous physical activity, as measured by thigh-positioned accelerometer (Axivity AX3®) worn for seven consecutive days. | Baseline, 18–20 months |
| | | Physical literacy | Between-group difference in change in total score of physical literacy as well as for the emotional, cognitive and physical subdomain. | Baseline, 18–20 months |
| | Biomarkers of intake and nutrient status | Whole-blood fatty acids, incl. n-3 LCPUFA | Between-group difference in change in whole-blood n-3 LCPUFA status, measured in fasting venous blood sample using high-throughput gas chromatography[a]. | Baseline, 6–8 months, (18–20 months)[a] |
| | | Blood hemoglobin | Between-group difference in change in whole-blood hemoglobin (iron status) (mmol/l), measured in fasting venous blood sample using auto-analyzer and immunoassay[a]. | Baseline, 6–8 months, (18–20 months)[a] |
| | | Serum ferritin | Between group difference in change in whole-blood ferritin (iron status) (ug/l), measured in fasting venous blood sample using auto-analyzer and immunoassay[a]. | Baseline, 6–8 months, (18–20 months)[a] |
| | | Vitamin D status (25-hydroxyvitamin D) | Between-group difference in change in serum 25-hydroxyvitamin D (vitamin D status) (nmol/l), measured in fasting venous blood sample using immunoassay or liquid chromatography-mass spectrometry[a]. | Baseline, 6–8 months, (18–20 months)[a] |
| | | Insulin-like growth factor-1 | Between-group difference in change in serum insulin-like growth factor-1 (growth factor) (ug/l), measured in fasting venous blood sample[a]. | Baseline, 6–8 months, (18–20 months)[a] |
| | Dietary intake and food literacy | Dietary intake | Between-group difference in change in dietary intake in g/day, as measured by 3-day dietary record (myfood24) and food frequency questionnaire (parent-reported). | Baseline, 6–8 months, 18–20 months |
| | | Food literacy | Between-group difference in change in food literacy using "GHK food literacy and meal culture questionnaire" (child-reported). | Baseline, 6–8 months, 18–20 months |
| | | School meal culture | Between-group difference in change in school meal culture using "GHK food literacy and meal culture questionnaire" (child-reported). | Baseline, 6–8 months, 18–20 months |
| | | Family meal culture | Between-group difference in change in family meal culture using "GHK food literacy and meal culture questionnaire" (child-reported). | Baseline, 6–8 months, 18–20 months |
| | Screen media practices | Total screen time | Between-group difference in change in total screen time (hours), as measured by modified SCREENS questionnaire (parent-reported). | Baseline, 6–8 months, 18–20 months |
| | | Smartphone/tablet use | Between-group difference in change in smartphone/tablet use, objectively assessed over 14 days using Ethica app. | Baseline, 6–8 months, 18–20 months |
| | Sleep time and quality | Total sleep time | Between-group difference in change in total sleep time (hours), as measured by thigh-positioned accelerometer (Axivity AX3®) worn for seven consecutive days. | Baseline, 18–20 months |
| | | Sleep quality | Between-group difference in change in sleep quality, as measured by Child Sleep Habits Questionnaire (parent-reported). | Baseline, 6–8 months, 18–20 months |

Abbreviations: BMI, Body mass index, CANTAB, Cambridge Neuropsychological Test Automated Battery; FM: Fat mass; FFM: Fat free mass; kg, kilograms; LCPUFA, long-chain polyunsaturated fatty acid; m, meters; mmHg, millimeters of mercury; mmol/l, millimole per liter; m/s, meters pr. Second; N, Newton; pmol/l, picomole per liter; s, second.

[a]Only children in Capital Region and Zealand Region. Measured at 18–20 months only if funding is available.

and sleep habits and -quality. Table 2 provides an overview of the pre-specified study outcomes which are also listed in www.clinicaltrials.gov (NCT05940675).

## Data management and data security

All collected data will be processed and stored in accordance with the EU General Data Protection Regulation (GDPR) as well as national Danish data protection legislation. The data collection is registered in the local records of data processing activities at the University of Southern Denmark, University of Copenhagen and the Capital Region of Denmark. Data collected via data entry (REDCap Mobile App) and surveys (links to REDCap surveys) will be stored directly in a secure database in REDCap, which is provided by the Odense Patient Data Explorative Network. Data validation settings (e.g. range checks for continuous variables) will be used to increase data quality. Data collected on external platforms or devices will be stored on restricted drives of the participating research institutions (e.g. heart rate data, accelerometry data) or company drives (MyFood24, Ethicadata, CambridgeCognition) with approved data processing agreements. Externally collected data will subsequently be uploaded to the REDCap database. Access to trial data will be restricted to research personnel working directly with data entry or analysis.

## Statistical analyses

The analyses of predefined primary and secondary outcomes will be planned à priori and made publicly available in a statistical analysis plan at www.clinicaltrials.gov (NCT05940675). The primary analysis will be conducted on an intention-to-treat basis using mixed regression analysis with school as random effect using degrees-of-freedom correction (Satterthwaite) to control for type-1 error due to the moderate number of clusters.

Analyses will be adjusted for baseline factors found to be related to individual schools (clusters) to control for possible baseline imbalance of cluster- and participant level data. In analyses using more than two assessment points, random slope effects of time will be employed to examine the possible heterogeneity of effects according to follow-up time. In all intention-to-treat analyses, data from all participating children at baseline will be included. A list of subgroup analyses will be predefined and outlined in the final statistical analysis plan at www.clinicaltrials.gov. All analyses that were not pre-planned will be labelled as exploratory. Missing values will be handled via the mixed regression modelling approach to analyses [91, 92]. If alternative analyses are carried out, missing values will be imputed using multiple imputation or inverse-probability weighting. All hypotheses testing will be based on two-sided tests at the alpha = 0.05 level.

In the food environment intervention, we will used linear mixed-effects regression to estimate the effect of the supermarket intervention on sales of selected food items, e.g. fruits and vegetables, in intervention stores compared with control stores. Index numbers will be used as the dependent variable which will be calculated as (weight of products sold in intervention period/weight of products sold same week in year before intervention) * 100, in line with prior studies [75–77].

## Study organization

The study is led by an Executive Board responsible for all strategic decisions affecting the conduct of the study. The Executive Board consists of the primary investigator (PI) (NBN), the co-primary investigator (JT) as well as UT, CTD, PK, AG, TS and RFK. The study is organized in six work packages (WPs): WP1 (led by UT) is responsible for coordination across work packages, establishment of national partnerships and local community capacity building. WP2 (led

by CTD), WP3 (led by PK) and WP4 (led by AG) will develop and deliver the interventions within Diet and Nutrition, Physical Activity, and Screen Media Use and Sleep, respectively. WP5 (led by RFK) is responsible for process and program evaluation, and WP6 (led by CTD) will coordinate measurements and data handling in the study. All work packages will employ postdocs and PhD students who will oversee contact and coordination with schools, data collection, and intervention delivery. The study also has an International Advisory Board consisting of internationally renowned researchers within child obesity prevention, and an Ethics Safety Board responsible for monitoring ethical and legal issues arising during the study. The project does not have an External Data Monitoring Committee due to the minimal risks associated with the intervention.

## Ethical considerations

The study was approved by the Regional Committee on Health Research Ethics for Southern Denmark (reference number S-20220094) on 14 April 2023 (protocol version 1.1) with subsequent amendments approved 6 July 2023 (version 2.1), 14 July 2023 (version 2.2) and 8 August 2023 (version 3.0, current version) (S3 File). Any subsequent protocol amendments will be submitted to the Health Research Ethics Committee for evaluation and will also be added to the registration at www.clinicaltrials.gov (NCT05940675). All custody holders of children participating in the study will provide written informed consent before enrolment of their child. Thorough participant information will be provided to custody holders and children, and all information to children will be given in a language and level appropriate for their age and comprehension. Participants in the biomedical measurement schedule are insured according to the Danish Act on Insurance for Work-related Injuries (No. 376, 31 Mar 2020) and Act on Access to Complaint and Compensation in the Health Care System (No. 995 of June 14, 2018).

The measurement procedures in the study are associated with minimal risk or discomfort for the participants. Staff conducting data collection will be trained to ensure that measurements are performed in a pleasant and safe environment, and great care will be taken to ensure that the children always feel comfortable and secure. Measurement days will be organized within the time frame of a normal school day, and there will be time for breaks to avoid exhausting the children. On measurement days, if a child declines participation or exhibits signs of discomfort or unease, the measurement will be discontinued, regardless of the custody holders' provision of informed consent. Potential discomfort associated with blood sampling will be minimized by use of local anesthetic patches, and the children will be offered breakfast immediately after the sample is taken. To avoid weight-related stigma and negative focus on body weight, results of the measurements will be concealed to the children. Furthermore, all communication about the study to children and parents will focus on wellbeing and general health, rather than a narrow focus on weight and obesity. All communication will be prepared carefully in terms of ensuring non-stigmatization of children or families. At intervention schools, all intervention components, including free school meals, extra physical activity during the school day, and family events, will be offered to all children and families regardless of whether the child participates in the biomedical measurement schedule.

## Adverse events, abnormal results and premature termination

The study's data manager will check the collected biomedical measurement data for potentially abnormal values using pre-specified clinical cutoffs. If outlying values are found, the clinically responsible physician (CM) will assess the result and determine whether the child's test result is clinically abnormal (e.g. hypertension). In this case, the child's parents will be informed.

Adverse events (AEs) will be monitored via questionnaires to parents at each measurement round, and all reported AEs will be assessed by the clinically responsible physician. If a serious AE occurs, the PI will notify the Regional Ethics Committee of Southern Denmark within seven days of becoming aware of such serious AE. The PI may temporarily suspend or prematurely terminate the project for significant and documented reasons.

## Publication and dissemination

The pre-registered primary and secondary outcomes of the trial will be reported following the CONSORT statement and published in a peer-reviewed journal. Results will also be presented at conferences to share the results with the scientific community and relevant stakeholders. Finally, lay summaries will be provided to the participating schools, local stakeholders, and parents of the participants on the project's website. Authors and contributors to publications related to the project will be defined following the recommendations of the International Committee of Medical Journal Editors (ICJME).

## Timeline and study status

Participant recruitment began on 14 August 2023, and the intervention will run for two school years from October 2023 to June 2025. Currently (February 2024), the intervention is ongoing, and the study has not yet generated results. Participant recruitment will continue throughout the intervention period (until 20 June 2025), as the study is an open cohort. The first follow-up assessment will occur from May–June 2024 (end of school year one), and second follow-up assessment will occur from May–June 2025 (end of school year two). Reporting of study results is expected before June 2026.

## Discussion

To our knowledge, the GHK trial is one of the first studies to combine a whole-systems approach to childhood obesity prevention with a rigorous cluster-randomized controlled evaluation design. Although a similar approach was applied in the WHO STOPS trial [26], a unique feature of GHK is the combination of fixed intervention elements which are pre-defined based on prior evidence (e.g., school lunch [59] and increased physical activity during school hours [60]) with systems thinking and co-created elements developed in collaboration with national and local stakeholders. The GHK program is ambitious as it incorporates interventions within four key behavioral areas (diet, physical activity, digital media habits and sleep) and targets multiple settings (families, schools, after-school clubs, local communities), in contrast to many prior studies focusing only on diet and physical activity in the school setting [13]. Furthermore, acknowledging the limitations of purely individual-level and educational strategies for health promotion [22, 27, 34], GHK focuses highly on changing the structural environments in which children and families spend their daily lives. As such, the GHK intervention represents a state-of-the-art, comprehensive, and coordinated program to prevent childhood obesity and promote overall child health and wellbeing.

An important aim of GHK is to reduce social inequalities in health and to reach families in socioeconomically disadvantaged positions. Several intervention elements are designed specifically to meet this goal, e.g., provision of free school lunches; working with after-school clubs to improve offers of healthy breakfast and after-school meals; organizing free-of-charge active summer camps for children with limited sporting experience; inviting parents and volunteers from local NGOs and social housing organizations to participate in the LCPGs; and working with local supermarkets to improve accessibility of healthy foods. Importantly, these

intervention components will be available to all families in intervention communities, regardless of whether their child participates in the biomedical measurement schedule.

A central challenge to the generalizability of the study findings relates to our ability to recruit participating schools. Almost 500 schools were invited, 24 agreed to participate, and one school withdrew participation after being randomized to the control group. The vast majority of invited schools did not respond to our invitation, but among those who responded, the most common cause for declining participation was lack of time and resources. This indicates that the participating schools are likely not representative of Danish schools in general, but rather represent a subset of schools with significant motivation and additional resources to invest in child health promotion. If GHK is found to be effective in these, presumably, highly motivated schools, adjustments to the intervention may be necessary to enable scale-up to Danish schools in general. In this respect, the comprehensive process evaluation of GHK will be crucial for identifying key contextual conditions necessary for sustainability and scale-up of the intervention.

A second potential challenge for the study relates to ensuring a high participation rate in the biomedical measurement schedule. In particular, since recruitment of study participants (i.e., children) will occur post-randomization, the participation rate may be lower at control schools. Furthermore, children who participate in the measurements may on average be healthier and have a higher socioeconomic position than non-participants, and this is likely to be more pronounced at control schools than intervention schools. This could lead to an underestimation of the potential intervention effect. To ensure the highest possible participation rate, we will collaborate closely with school principals and teachers on how to best reach the eligible families, and members from the research team will continuously visit schools and after-school clubs during the recruitment period to answer any questions and concerns from parents. Furthermore, pseudonymized register data will be obtained on the socioeconomic position of both participating and non-participating children [89], which will enable us to estimate the likely effect of non-participation on our results.

With respect to the potential for sustainability and further scale-up of GHK after the end of the intervention period, several intervention components are designed specifically to support sustainability of the intervention at participating schools: Firstly, most intervention components will be delivered by teachers and after-school staff, rather than external research staff, and some intervention elements (e.g., physical activity) will be integrated into the regular school curriculum. Secondly, the parallel work to adapt the program further to local conditions, which is especially planned for the second year of implementation, will help ensure that core intervention activities are anchored in the school's usual practice. Finally, the local community intervention will focus on building organizational capacity in local municipalities and establishing local partnerships and networks which is known to be crucial for intervention sustainability [93]. Also, if the GHK intervention is found to be effective, the national and local partnerships established as part of the study (including collaborations with national government agencies such as the Danish Health Authority and the Danish Food and Veterinary Administration) will help facilitate the possibility for wider scale-up of the intervention to regional or national level.

## Supporting information

**S1 File. Completed SPIRIT checklist.**
(PDF)

**S2 File. Additional supporting information, including Overview of municipalities included in each school invitation round (S1 Table), Overview of requirements for schools to**

**participate (S2 Table), Details of Generation Healthy Kids Intervention components (S3-S5 Tables), and Detailed description of measurement procedures.**
(PDF)

**S3 File. Protocol version 3.0 approved by the Regional Ethics Committee of Southern Denmark on 8 August 2023.**
(PDF)

**S4 File. Completed TIDieR checklist.**
(PDF)

## Acknowledgments

The authors gratefully acknowledge all staff in the Generation Healthy Kids Study Group which at the time of submission include: Anna Eilersen (WP2), Anne Lykke Poulsen (study secretariat), Bat-El Menadeva Karpantschof (WP2, WP6), Bo Kousgaard Poulsen (communication team), Danielle Nørager Johansen (WP1), Dorte William Wedell-Wedellsborg (WP1), Frederik Holmegaard Jensen (WP2, WP6), Ida Foxvig (WP1), Inge Rasmussen (WP2), Jane Jørgensen (WP2), Jonas Vestergaard Nielsen (WP1), Kristian Levring Madsen (communication team), Kristian Overgaard (WP3), Kristine M. Kristensen (WP2), Lene Stevner (Good Clinical Practice coordinator), Line Kattai Ulrikkeholm (WP2), Louise A. S. Brautsch (WP5), Louise Stjerne Madsen (communication team), Maja Vilhelmsen (WP1), Maja Sulstad Johansen (WP4), Mette Lindholm Kurtzhals (WP3), Pia Sandfeld Melcher (WP3), Sofie Koch (WP1), and Tine Buch Andersen (WP1). We also acknowledge all student assistants and scientific interns assisting with data collection. Furthermore, we acknowledge the members of our International Scientific Advisory Board for important scientific input to the design and conduct of the study: Professor Emeritus Adrian Bauman (University of Sydney, Australia); professor Harry Rutter (Department of Social & Policy Sciences, University of Bath, United Kingdom), professor Carolyn Summerbell (Department of Sport and Exercise Sciences, Durham University, United Kingdom), professor Steve Gortmaker (Department of Social and Behavioral Sciences, Harvard School of Public Health, USA), professor Frøydis N. Vik (Department of Nutrition and Public Health, University of Agder, Norway), and professor Joan L. Duda (School of Sport, Exercise and Rehabilitation Sciences, University of Birmingham, United Kingdom).

## Author Contributions

**Conceptualization:** Camilla Trab Damsgaard, Peter Krustrup, Anders Grøntved, Rikke Fredenslund Krølner, Glen Nielsen, Jesper Lundbye-Jensen, Thomas Skovgaard, Christian Mølgaard, Jens Troelsen, Nikolai Baastrup Nordsborg, Ulla Toft.

**Data curation:** Jesper Schmidt-Persson, Malte Nejst Larsen, Paulina Sander Melby, Natascha Holbæk Pedersen.

**Formal analysis:** Louise T. Thomsen, Jesper Schmidt-Persson, Camilla Trab Damsgaard, Peter Krustrup, Anders Grøntved, Glen Nielsen, Jesper Lundbye-Jensen, Malte Nejst Larsen, Paulina Sander Melby, Natascha Holbæk Pedersen.

**Funding acquisition:** Camilla Trab Damsgaard, Peter Krustrup, Anders Grøntved, Rikke Fredenslund Krølner, Glen Nielsen, Jesper Lundbye-Jensen, Thomas Skovgaard, Christian Mølgaard, Jens Troelsen, Nikolai Baastrup Nordsborg, Ulla Toft.

**Investigation:** Louise T. Thomsen, Jesper Schmidt-Persson, Anders Blædel Gottlieb Hansen, Didde Hoeeg, Malte Nejst Larsen, Line Lund, Paulina Sander Melby, Natascha Holbæk Pedersen.

**Methodology:** Camilla Trab Damsgaard, Peter Krustrup, Anders Grøntved, Rikke Fredenslund Krølner, Glen Nielsen, Jesper Lundbye-Jensen, Thomas Skovgaard, Christian Mølgaard, Anders Blædel Gottlieb Hansen, Jens Troelsen, Nikolai Baastrup Nordsborg, Ulla Toft.

**Project administration:** Louise T. Thomsen, Jesper Schmidt-Persson, Camilla Trab Damsgaard, Peter Krustrup, Anders Grøntved, Rikke Fredenslund Krølner, Didde Hoeeg, Malte Nejst Larsen, Line Lund, Natascha Holbæk Pedersen, Jens Troelsen, Nikolai Baastrup Nordsborg, Ulla Toft.

**Supervision:** Jesper Schmidt-Persson, Camilla Trab Damsgaard, Peter Krustrup, Anders Grøntved, Rikke Fredenslund Krølner, Glen Nielsen, Jesper Lundbye-Jensen, Thomas Skovgaard, Didde Hoeeg, Malte Nejst Larsen, Line Lund, Jens Troelsen, Nikolai Baastrup Nordsborg, Ulla Toft.

**Writing – original draft:** Louise T. Thomsen, Jesper Schmidt-Persson.

**Writing – review & editing:** Camilla Trab Damsgaard, Peter Krustrup, Anders Grøntved, Rikke Fredenslund Krølner, Glen Nielsen, Jesper Lundbye-Jensen, Thomas Skovgaard, Christian Mølgaard, Anders Blædel Gottlieb Hansen, Didde Hoeeg, Malte Nejst Larsen, Line Lund, Paulina Sander Melby, Natascha Holbæk Pedersen, Jens Troelsen, Nikolai Baastrup Nordsborg, Ulla Toft.

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
