## [Decision Letter · Decision Letter 0]

29 May 2024

PONE-D-24-05389Generation Healthy Kids: Protocol for a cluster-randomized controlled trial of a multi-component and multi-setting intervention to promote healthy weight and wellbeing in 6-11-year-old children in DenmarkPLOS ONE

Dear Dr. Thomsen,

Thank you for submitting your manuscript to PLOS ONE. After careful consideration, we feel that it has merit but does not fully meet PLOS ONE’s publication criteria as it currently stands. Therefore, we invite you to submit a revised version of the manuscript that addresses the points raised during the review process.

We look forward to receiving your revised manuscript.

Kind regards,

Maiken Pontoppidan

Academic Editor

PLOS ONE

Journal Requirements:

3.Please review your reference list to ensure that it is complete and correct. If you have cited papers that have been retracted, please include the rationale for doing so in the manuscript text, or remove these references and replace them with relevant current references. Any changes to the reference list should be mentioned in the rebuttal letter that accompanies your revised manuscript. If you need to cite a retracted article, indicate the article’s retracted status in the References list and also include a citation and full reference for the retraction notice.

Additional Editor Comments (if provided):

Dear authors,

Thank you for submitting your manuscript entitled “Generation Healthy Kids: A Multi-Setting, Multi-Component Intervention to Promote Healthy Weight Development in Danish Children Aged 6–11 Years” to PLOS ONE.

Your study protocol is well-written and outlines an impressive and comprehensive approach to addressing childhood obesity. The design of the GHK study, utilizing a cluster-randomized trial across multiple schools and incorporating diverse settings, is robust and promising.

The emphasis on capacity building and stakeholder involvement is particularly noteworthy, offering a novel strategy to promote child health and prevent obesity.

Reviewer 1 has provided some valuable comments and suggestions that can further strengthen your manuscript. I recommend that you carefully consider and incorporate these comments in your revision.

I only have a few minor comments (all are mentioned by reviewer 1 as well):

L 389 Consent: Do you obtain consent from one or both if there is shared custody?

L441: will all children get the free lunch irrespective of they participate in the study or not?

L 640 data: is it voluntary for the children? Can they decide to not do the tests on the assessment days if there is consent from parents?

Thank you for your valuable contribution to the field. I look forward to receiving your revised manuscript.

Reviewers' comments:

Reviewer's Responses to Questions

**Comments to the Author**

1. Does the manuscript provide a valid rationale for the proposed study, with clearly identified and justified research questions?

Reviewer #1: Yes

2. Is the protocol technically sound and planned in a manner that will lead to a meaningful outcome and allow testing the stated hypotheses?

Reviewer #1: Partly

3. Is the methodology feasible and described in sufficient detail to allow the work to be replicable?

Reviewer #1: Yes

4. Have the authors described where all data underlying the findings will be made available when the study is complete?

Reviewer #1: Yes

5. Is the manuscript presented in an intelligible fashion and written in standard English?

Reviewer #1: Yes

6. Review Comments to the Author

You may also provide optional suggestions and comments to authors that they might find helpful in planning their study.

Reviewer #1: Thank you for the opportunity to review this paper, which addresses the crucial topic of childhood obesity. The authors have outlined a protocol for a comprehensive intervention targeting children aged 6-11 and its implementation from the children's perspective.

Overall, the paper is well-written, and the study and intervention are clearly described. However, it would benefit from adhering to the CONSORT guidelines to enhance its rigor. While the authors plan to use CONSORT when publishing the results, the protocol itself would also benefit from following these guidelines. The intervention description is generally clear but could be improved by using a standardized framework such as the TIDieR tool. I have some specific comments and suggestions for improving the manuscript.

Abstract: The sample size should be mentioned in more detail, including a justification (power analysis) for the targeted sample size. The abstract also lacks a detailed description of the outcomes, measurement time points, and focuses more on the intervention itself. Additionally, the timing of the study implementation is missing.

Background: The flow is good, and the intervention and its goals, as well as the state of the art, are well-described. Starting at line 145, the intervention description begins. Several approaches form the basis for the intervention, such as the population-based primary prevention approach (line 147) and the whole-system approach (line 190), but there is no information about what these approaches entail or the references they are based on. Readers would benefit from a few sentences elaborating on these approaches. Since the intervention aims for behavioral changes, it would be helpful to describe the theoretical framework underpinning the intervention. Terms like "lower" and "most" in lines 160 and 174 should be accompanied by precise numbers to avoid vagueness, as seen with the "10%" mentioned in line 185. The sentence in lines 188-190 is complex and needs clarification, particularly about who the stakeholders are. Additionally, the paper lacks a cohesive description of the intervention's development, as it is scattered across different sections, making it difficult to follow. Although the pilot and feasibility study (line 199) will be reported in a separate paper, it still needs further explanation regarding sample sizes, what was measured, and when and where these measurements took place. The paragraph starting at line 209 includes a list-like text, and some descriptions would be better placed in the methods section, especially the list of outcomes. From line 230, if the authors are following a framework for implementation science, it should be mentioned to justify the measures taken to evaluate the intervention's implementation.

Design and Methods: As previously mentioned, the methods section would benefit from being structured according to CONSORT guidelines. In line 250, please specify the eligibility criteria for the schools. In line 319, clarify the requirements for the schools. Regarding line 389, was there a reason for not seeking consent from the children? While parental consent is necessary, children are capable of providing their own informed consent. This issue should be addressed and elaborated upon in the ethics section, including how researchers ensure the willingness and voluntary participation of the children.

In lines 407 and 409, the authors mention core elements and co-creation elements. However, in the detailed intervention description that follows, the distinction between these elements is not consistently mentioned. A detailed table outlining these elements, including their frequency, timing (e.g., whether they last the entire two years and start from the beginning), target groups, and other critical aspects of their implementation, would help readers better understand the complex intervention. If not presented in a table, the text should include these important details to demonstrate the rigor of the intervention.

Line 419 discusses the development of the intervention and would be better placed in the section detailing the development process, prior to the methods section. A process graph illustrating all phases of the development process would also be beneficial. Additionally, it is important to mention if children, parents, or teachers were included in the development process, as this participatory approach is crucial in designing and developing health interventions.

From line 440, it is unclear what happens to children who do not participate in the study. Will they still receive the intervention? It is important to mention that non-participating children will not be treated unequally. The section from lines 482-486 is brief and needs more elaboration on the content of this element. For example, lines 534 and 553 describe possible differences between interventions in different contexts. How will the authors ensure that the results are not affected by these differences if the intervention is not fully standardized? From line 538, it is unclear when the children's visions are communicated to other stakeholders and what actions will follow.

Overall, there are instances where the authors describe actions performed by the research group and refer to "us." Are "we" the same as "the research group"? This should be clarified. From line 714, explain how the effects of the food environment intervention are calculated. From line 722, provide more details about the field observations, focus groups, interviews, and surveys, including what is evaluated, the sample size, and the timing of these evaluations.

Ethical Considerations: As mentioned regarding the consent issue, the section also needs a more detailed reflection on the study's impact on children. The numerous measurements and questionnaires might be exhausting for the children, and this should be considered and addressed.

Discussion: In line 868, it is unclear how the co-creation elements were developed and implemented. More detailed information on the co-creation process is needed. This should include how stakeholders, including children, parents, teachers, and other relevant parties, were involved in the co-creation process, and the specific steps taken to ensure their meaningful participation.

Thank you for the opportunity to review your paper. I wish you all the best in implementing your intervention. Your work on addressing childhood obesity is vital and commendable, and I look forward to seeing the results of this important study.

7. PLOS authors have the option to publish the peer review history of their article (what does this mean?). If published, this will include your full peer review and any attached files.

Reviewer #1: No

---

## [Author Response · Author response to Decision Letter 0]

4 Jul 2024

Please see our detailed response to comments from the Editor and Reviewer in the attached file "Response to reviewer and editor comments".

---

## [Editor Report · Decision Letter 1]

18 Jul 2024

Generation Healthy Kids: Protocol for a cluster-randomized controlled trial of a multi-component and multi-setting intervention to promote healthy weight and wellbeing in 6-11-year-old children in Denmark

PONE-D-24-05389R1

Dear Dr. Thomsen,

We’re pleased to inform you that your manuscript has been judged scientifically suitable for publication and will be formally accepted for publication once it meets all outstanding technical requirements.

Kind regards,

Maiken Pontoppidan

Academic Editor

PLOS ONE

Additional Editor Comments (optional):

Dear authors

Thank you for resubmitting the manuscript titled "Generation Healthy Kids: Protocol for a Cluster-Randomized Controlled Trial of a Multi-Component and Multi-Setting Intervention to Promote Healthy Weight and Wellbeing in 6-11-Year-Old Children in Denmark".

Your thorough revisions and detailed responses to the reviewers' comments have significantly enhanced the quality and clarity of the paper. We appreciate your efforts in addressing the feedback and improving the manuscript.

Thank you for your valuable contribution to the field.
---

## [Editor Report · Acceptance letter]

13 Aug 2024

PONE-D-24-05389R1 

PLOS ONE

Dear Dr. Thomsen, 

I'm pleased to inform you that your manuscript has been deemed suitable for publication in PLOS ONE. Congratulations! Your manuscript is now being handed over to our production team.

Kind regards, 

on behalf of

Dr. Maiken Pontoppidan 

Academic Editor

PLOS ONE